# Synthesis of AZO-Coated ZnO Core–Shell Nanorods by Mist Chemical Vapor Deposition for Wastewater Treatment Applications

**DOI:** 10.3390/molecules29020309

**Published:** 2024-01-08

**Authors:** Htet Su Wai, Tomoya Ikuta, Chaoyang Li

**Affiliations:** 1School of Systems Engineering, Kochi University of Technology, 185 Miyanokuchi, Tosayamada Cho, Kami City 782-8502, Kochi, Japan; htet.su.wai@kochi-tech.ac.jp (H.S.W.);; 2Center of Nanotechnology, Kochi University of Technology, 185 Miyanokuchi, Tosayamada Cho, Kami City 782-8502, Kochi, Japan

**Keywords:** AZO-coated ZnO core–shell nanorods, zinc oxide nanorods, aluminum-doped zinc oxide seed layer, mist chemical vapor deposition, photodegradation efficiency

## Abstract

AZO-coated ZnO core–shell nanorods were successfully fabricated using the mist chemical vapor deposition method. The influence of coating time on the structural, optical, and photocatalytic properties of zinc oxide nanorods was investigated. It was observed that the surface area of AZO-coated ZnO core–shell nanorods increased with an increase in coating time. The growth orientation along the (0001) crystal plane of the AZO thin film coating was the same as that of zinc oxide nanorods. The crystallinity of AZO-coated ZnO core–shell nanorods was significantly improved as well. The optical transmittance of AZO-coated ZnO core–shell nanorods was greater than 55% in the visible region. The degradation efficiency for methyl red dye solution increased with an increase in coating time. The highest degradation efficiency was achieved by AZO-coated ZnO core–shell nanorods with a coating duration of 20 min, exhibiting a degradation rate of 0.0053 min^−1^. The photodegradation mechanism of AZO-coated ZnO core–shell nanorods under ultraviolet irradiation was revealed.

## 1. Introduction

In recent decades, industrialization has been a driving force behind the progress of society. However, it has also posed a significant challenge: the need to reduce pollution and limit the consumption of fossil fuels [1,2,3]. Annually, an estimated 7 × 10^5^ tons of organic dyes are released globally, with over 16% of this amount being discharged into wastewater [4,5,6,7]. Among the various organic dyes in wastewater, methyl red (MR), an azo dye recognized for its harmful and toxic effects on human health, is a matter of specific concern. MR and similar organic dyes are extensively utilized in various industries, including textiles and pharmaceuticals, and their discharge into wastewater can lead to severe environmental contamination [8,9,10]. MR is known to be harmful and toxic to aquatic life, posing a significant threat to the delicate balance of aquatic ecosystems. Moreover, MR can persist in water bodies for extended periods, causing long-term environmental harm and potentially entering the food chain, which can have detrimental health effects on human health and wildlife. Removing MR from wastewater is crucial for protecting the environment, preserving biodiversity, and safeguarding the health and well-being of ecosystems and communities that depend on clean water sources [11,12,13]. The degradation of the MR dyes involves the catalytic activity of semiconductor materials under photon irradiation. Various metal oxides used as semiconductor photocatalysts, including zinc oxide (ZnO), titanium dioxide, copper oxide, silver oxide, and iron (III) oxide, have been applied in the degradation of organic dyes [14,15,16,17]. Among various photocatalyst materials, nanostructured ZnO thin films have garnered significant attention due to their unique properties, including a direct wide bandgap (3.37 eV), large exciton binding energy (60 meV) at room temperature, non-toxicity, chemical stability, and low cost. Because the photodegradation efficiency of ZnO is greatly influenced by its surface morphology, modifications of ZnO nanostructures, such as spheres, rods, tubes, and needles, play a pivotal role in improving photodegradation performance.

Recently, one-dimensional ZnO nanorods have attracted much attention as photocatalyst materials in degrading MR dyes and removing pollutants from wastewater environments due to their high adsorption of MR dyes and superior photodegradation efficiency [18]. Notably, in the growth of ZnO crystals, the orientation along the (0001) crystal plane demonstrates remarkable activity in photodegradation because of its high surface energy and polar character. This polar facet enables efficient degradation of organic contaminants, thus improving photocatalytic efficiency [19,20,21,22]. Additionally, controlling the growth of large surface area is crucial, as it enhances the ability to absorb chemical reactants from the degradation process, facilitating the high production of reactive oxygen species (ROS) and improving the photodegradation efficiency. In our previous research [23], we achieved the fabrication of well-aligned ZnO nanorods with high crystallinity and enlarged surface area on aluminum (Al)-doped ZnO (AZO) substrates using the chemical bath deposition (CBD) method. However, the photocatalytic degradation rate was limited.

The high recombination rate of photogenerated electrons and holes in ZnO prevents their effective involvement in the degradation of MR, resulting in a decrease in photocatalytic efficiency. Strategies to minimize recombination, such as surface modifications, are essential. The reduction in the recombination rate of electron–hole pairs in ZnO has been addressed through surface modification, introduction of defects, or doping with transition metals. In contrast to alternative methodologies, the incorporation of Al is expected to represent a more efficient approach for decreasing the recombination rate of electron–hole pairs in ZnO. When Al atoms replace some of the zinc atoms in the ZnO crystal lattices, additional energy levels within the band structure are introduced. The introduction of these aluminum impurity states results in a change in the electronic structure of the material, leading to an increase in the bandgap [24]. A large bandgap can enhance the separation of charge carriers, reducing the recombination rate of electron–hole pairs. In our previous research [25], Al was effectively doped in ZnO thin film with precise and controllable doping ratios by using the mist chemical vapor deposition (CVD) method, resulting in enlarged bandgap energy. However, the resultant low crystallinity of AZO film remained a subject of concern. To improve the degradation efficiency of ZnO, it is crucial to simultaneously mitigate the recombination rate of ZnO while maintaining its high crystalline quality and substantial surface area.

In this research, we propose to combine the both CBD and mist CVD methods for the fabrication of AZO-coated ZnO (AZO/ZnO) core–shell nanorods. The obtained AZO/ZnO core–shell nanorods are expected to reduce the recombination rate of charge carriers efficiently, thereby enhancing the photodegradation efficiency. The structural, optical, and photocatalytic properties of AZO/ZnO core–shell nanorods will be investigated.

## 2. Results and Discussion

Figure 1A shows the SEM images of the 300 nm thick AZO film deposited by RF magnetron sputtering and ZnO nanorods grown on the as-deposited AZO film using the CBD method. In the top view image, a uniformly and flat AZO film with an average grain size of 53 nm can be observed. The columnar structures of AZO film showed vertical growth on the glass substrate in the cross-section view, as shown in Figure 1(A1-b). After the 5h CBD process, the obtained ZnO nanorods exhibited a well-defined hexagonal shape which was grown vertically with a length of 1173 nm. The growth direction of obtained ZnO nanorods followed a high alignment with that of the as-deposited AZO film, as shown in Figure 1(A2-a,A2-b).

The XRD patterns of the as-deposited AZO film and ZnO nanorods synthesized on AZO film are shown in Figure 1B. It was observed that there was only (002) diffraction peak for the as-deposited AZO film, which meant that the preferred growth direction was in the c-axis growth orientation along the (0001) crystal plane. After the CBD process, still only the (002) peak was observed, which revealed that the growth direction of ZnO nanorods followed the same growth direction of the as-deposited AZO film. The crystallinity of ZnO nanorods was much improved compared to that of the AZO film.

The SEM images of AZO/ZnO core–shell nanorods with different AZO coating times using the mist CVD process are shown in Figure 2. From the top views, the obtained AZO/ZnO core–shell nanorods exhibited an almost-circular structure, and the diameters of the AZO/ZnO core–shell nanorods were increased from 165 nm to 355 nm as the coating time was extended from 5 min to 20 min. The result revealed the average diameter of ZnO nanorods was increased with the increase in AZO coating time.

Compared to the hexagonal wurtzite facets for ZnO nanorods with an average diameter of 104 nm before the coating process, the surfaces of ZnO nanorods gradually grew and transformed into an almost-circular shape with an increase in AZO coating time. The surface morphology change might be due to the difference in ionic radius between Zn^2+^ (0.74 Å) and Al^3+^ (0.53 Å) [26]. When the AZO was coated on the ZnO nanorods during the mist CVD process, the high concentration of Al ions might replace the Zn ion sites, resulting in structural changes. Moreover, it can be determined that the thicknesses of AZO-coated layers on ZnO nanorods increased from 42 nm to 137 nm when the coating time was extended from 5 min to 20 min. From the cross-sectional views, it was observed that small AZO nanoparticles attached to the ZnO nanorods after coating for 5 min and then gradually became a thin film covering the ZnO nanorods uniformly as the coating time increased. When the coating time reached 20 min, all the ZnO nanorods were successfully coated with AZO, and the diameter increased four times. The largest average diameter of 355 nm was obtained for AZO/ZnO core–shell nanorods with 20 min-coating, while the length of the ZnO nanorods had little change.

An EDX measurement was carried out in order to evaluate the elemental composition of the AZO/ZnO core–shell nanorods with different coating times during the mist CVD process. The variations in the atomic percentages of Al/Zn, (Al + Zn)/O, Zn/O, and Al/O are shown in Figure 3. Compared to the non-coated ZnO nanorods, higher atomic ratios of Al/Zn and (Al + Zn)/O from the AZO/ZnO core–shell nanorods were observed, which indicated that much more Al atoms were successfully incorporated onto the ZnO nanorods during the AZO coating using the mist CVD process. It was clearly observed that the atomic ratios of Zn/O and Al/O were significantly increased when the AZO coating time was extended from 5 min to 20 min. The Zn/O ratio was much higher than that of Al/O, indicating that the Zn-O bond might be enhanced with an increase in coating time, resulting in crystallinity improvement with the increase in AZO coating time.

Figure 4A shows the XRD patterns of AZO/ZnO core–shell nanorods with different coating times using the mist CVD method. It was observed that the (002) peak was the dominant peak for AZO/ZnO core–shell nanorods with different coating times, which meant that the obtained AZO/ZnO core–shell nanorods highly preferred c-axis orientation growth along the (0001) crystal plane. When the AZO coating time was extended to 20 min, the (101) crystal plane appeared due to a high concentration of Al atoms replaced the sites of Zn, however the (002) diffraction peak was still dominant and exhibited the highest crystallinity.

The analysis from XRD measurement is shown in Figure 4(B1). The intensity of the (002) diffraction peak was increased, and the full width at half maximum value of the (002) diffraction peak decreased as the coating time was extended from 5 min to 20 min, which indicated that the crystallinity of the AZO/ZnO core–shell nanorods was improved as the coating time increased. Based on Scherrer’s formula [27], the calculated average c-axis crystallite size of the AZO/ZnO core–shell nanorods increased from 31.2 nm to 36.4 nm, indicating that growth in the c-axis direction was highly preferred. According to the biaxial strain model [28], the compressive stress (σ) can be determined as follows:(1)$$\sigma =\frac{2C_{13}^{2}-C_{33}(C_{11}+C_{12})}{2C_{13}}\;\times \;\frac{C_{film}-C_{bulk}}{C_{bulk}}$$

Here, C is the lattice constant, *C_ij_* is the elastic modulus of bulk ZnO films, *C*_11_ = 208.8 GPa, *C*_12_ = 119.7 GPa, *C*_13_ = 104.2 GPa, and *C*_33_ = 213.8 GPa. The peak position of (002) was right-shifted from 34.39°, 34.40°, 34.43° to 34.45° with the increase in AZO coating time, and the compressive stress was reduced from −0.984 GPa to −0.455 GPa when the AZO coating time was increased from 5 min to 20 min, as shown in Figure 4(B2).

In order to investigate the influence of crystal orientation ZnO nanorods contributing to the growth direction of AZO layers on ZnO nanorods during the mist CVD process, the lattice mismatch ratios between the AZO/ZnO core–shell nanorods and non-coated ZnO nanorods were calculated based on the lattice mismatch theory.
(2)$$\mathrm{Lattice}\;\mathrm{mismatch}\;\mathrm{ratio}\;(\%)=\frac{C_{ZnO\;NRs}-C_{AZO/ZnO}}{C_{ZnO\;NRs}}\;\times \;100\%$$

The calculated mismatch ratios between non-coated ZnO nanorods and AZO/ZnO core–shell nanorods decreased from 0.25% to 0.038% as the AZO coating time was extended from 5 min to 20 min. The lowest lattice mismatch ratio was attained in AZO/ZnO core–shell nanorods with a 20 min-AZO coating time. Therefore, it is confirmed that the crystal orientation of non-coated ZnO nanorods played a contributory role in directing the growth orientation along the (0001) crystal plane of the AZO film during the mist CVD process.

Figure 5a demonstrates the model of the growth process of the AZO/ZnO core–shell nanorods at 400 °C during the mist CVD method. The mechanism of growth of AZO film using the mist CVD process was already revealed in our previous research [25,29]. As shown in Figure 5a, the generated mist droplets, including zinc acetate (ZA)and aluminum acetylacetonate (AA), were transported to the reaction chamber, in which ZnO nanorods were heated and kept at 400 °C. During the AZO coating process, the transported mist droplets could be decomposed and release Al and Zn ions on the ZnO nanorods; the AlO_6_ bonding mode could occur when the deposition temperature was over 350 °C during the mist deposition process, and the released Al ions could be bonded with oxygen and form Al_x_O_1−x_. Meanwhile, the decomposed Zn ions from mist droplets were also bonded with oxygen to form ZnO. Based on the previous EDX analysis, the Zn-O bonding were dominatedly, which led to the formation of more ZnO nuclei on the surface of the ZnO nanorods. According to the minimum energy principle of ZnO along the (0001) growth direction, the growth direction of the obtained AZO films could follow the same growth direction as the ZnO nanorods, thus enhancing the crystallinity along the (0001) crystal plane.

In addition, because the mist CVD deposition temperature was set at 400 °C, the influence of thermal annealing on the crystallinity of as-deposited ZnO nanorods before AZO coating was also investigated. In order to clarify the influence of crystallinity on the AZO/ZnO core–shell nanorods, XRD patterns of ZnO nanorods, ZnO nanorods annealed at 400 °C for 20 min, and AZO/ZnO core–shell nanorods were compared, as shown in Figure 5b. Based on the XRD results, it can be clearly observed that the intensity of the (002) diffraction peak of annealed ZnO nanorods was nearly same as that of as-deposited ZnO nanorods, which meant that the thermal annealing had little effect on the crystallinity of ZnO nanorods during the mist CVD process.

Figure 6a shows the transmission spectra of AZO/ZnO core–shell nanorods with different AZO coating times using the mist CVD method. The transmittance was gradually decreased from 71% to approximately 55% as the AZO coating time was extended from 5 min to 20 min during the mist CVD process.

The optical bandgap variations of the AZO/ZnO core–shell nanorods, as determined by plotting (*αhv*)^2^ versus the photon energy (*hv*), with different AZO coating times are presented in Figure 6b. Tauc’s plot equation [30] was applied to calculate the optical bandgap energy of the AZO/ZnO core–shell nanorods, and it is expressed as follows:(*αhv*)^2^ = *A* (*hv* − *E_g_*)(3)
where α is absorption coefficient, *h* is Planck’s constant, *v* is the photon frequency, *A* is a constant, and *E_g_* is the optical bandgap. According to the calculated results, the optical bandgaps of the AZO/ZnO core–shell nanorods were enlarged from 3.52 eV to 3.65 eV when the coating time was increased from 5 min to 20 min. As is well known, in the recombination process of photogenerated charge carriers, a large bandgap is associated with elevated photon energy. The presence of the large bandgap in the obtained AZO/ZnO core–shell nanorods could result in a greater energy differential between the conduction and valence bands, thereby reducing the recombination rate of charge carriers. In this study, the increased bandgap energy might contribute to suppressing the recombination rate of electrons and holes during the degradation process.

Photodegradation measurement was carried out after coating AZO on ZnO nanorods with different timesThe absorption spectra of the MR solution for both the as-deposited ZnO nanorods and the AZO/ZnO core–shell nanorods with different AZO coating times using the mist CVD method under UV irradiation are shown in Figure 7a. In order to compare the degradation properties after irradiation, the highest absorption band of MR was selected at 520 nm which is pointed in Figure, which is a basic form of MR dye under irradiation [31]. With the extension of the AZO coating times from 5 min to 20 min, a significant reduction in absorption intensity at 520 nm compared to the original MR solution was clearly observed. This phenomenon could be attributed to the high generation of OH^−^ radicals from the AZO/ZnO core–shell nanorods, which enhanced their capability to interact with MR dye molecules. Consequently, this increased interaction led to a more extensive breakdown of MR chains, which contributed to the observed decrease in absorption intensity at around 520 nm. The AZO/ZnO core–shell nanorods with 20 min-AZO coating showed the lowest absorption intensity, indicating the highest photodegradation efficiency.

The MR dye fading of AZO/ZnO core–shell nanorods with different AZO coating times is described in Figure 7b. It was found that the color of the original MR dyes gradually changed from red to pale colors after the degradation process with the AZO coating time increasing, indicating that the degradation efficiency of MR dyes was improved with the increase in AZO coating time. An almost-transparent color after degradation was achieved using AZO/ZnO core–shell nanorods with 20 min-AZO coating.

In general, the electrons (e^−^) of ZnO nanorods from the valance band were excited to the conduction band and then left holes (h^+^) within the valance band upon UV light exposure. The photogenerated e^−^ could be easily moved on the surface of ZnO nanorods and oxidized with O_2_ to form superoxide radicals (${.O}_{2}^{-}$). Simultaneously, h^+^ could react with water molecules (H_2_O) to release OH^−^ ions. The generated ${.O}_{2}^{-}$ continued to react with decomposed hydrogen ions (H^+^) created from H_2_O, leading to the formation of H_2_O_2_. This sequence of reactions culminates in the creation of °OH radicals, which can degrade the MR dyes efficiently [32,33,34]. There might be three reasons for the high degradation efficiency achieved using the obtained AZO/ZnO core–shell nanorods in this experiment. Firstly, the obtained AZO/ZnO core–shell nanorods with high crystallinity possess (0001) facets, which have a large surface and abundant active sites for generating many °OH radicals to increase the production rate of ROS under UV light irradiation. Secondly, the optical bandgap energy was increased as the AZO coating time was increased from 5 minutes to 20 min, corresponding to the enhanced absorption of energy photons and increasing the number of charge carriers generated. Third, the bandgap of the AZO/ZnO core–shell nanorods was enlarged with an increase in coating time, suppressing the recombination rate of electron–hole pairs.

The changes in the absorption spectra of MR solution provide insight into the photocatalytic degradation rates, as described in Table 1. The Langmuir–Hinshelwood (L-H) model [35] can be applied to calculate AZO/ZnO core–shell nanorods degradation kinetics. The adsorption rate can be expressed through the coverage ratio of reactants adsorbed onto the surface of a photocatalyst. The kinetic rate of the degradation reaction, represented as K, can be mathematically described by the following equation:k = ln (C_t_/C_o_)/t (4)

The obtained first-order degradation reaction rate (k) is calculated using the initial dye concentration (Co) and the dye concentration at a specific time (C_t_). Based on the L-H model, a higher first-order degradation reaction rate indicates a high photodegradation efficiency. Under UV light irradiation, the first-order degradation reaction rate for AZO/ZnO core–shell nanorods was increased from 0.0039 min^−1^ to 0.0042 min^−1^, 0.0043 min^−1^, and 0.0053 min^−1^ as the AZO coating time was extended from 5 min to 20 min. Notably, the highest degradation rate of MR dye solution was observed for AZO/ZnO core–shell nanorods with an AZO coating time of 20 min.

## 3. Experiments

### 3.1. Deposition of AZO Film by Radio Frequency Magnetron Sputtering

A 300 nm thick AZO film (ZnO:Al_2_O_3_ = 98:2 (wt%)) was deposited onto an alkaline-free glass substrate (Eagle XG) using a radio frequency (RF) magnetron sputtering system, which operated at a frequency of 13.56 MHz. Before sputtering, the glass substrate was pre-heated at 150 °C for one hour. Pure argon gas with a flow rate of 30 sccm was introduced into the chamber as the working gas. The pressure was kept at 1 Pa, while the temperature was consistently maintained at 150 °C during the deposition process. The deposition conditions of AZO film are shown in Table 2.

### 3.2. Synthesis of ZnO Nanorods by CBD

Following the deposition process, ZnO nanorods were synthesized on as-deposited AZO film utilizing the CBD method. Throughout the CBD process, a mixed precursor solution consisted of Zn (NO_3_)_2_·6H_2_O with a concentration of 0.015 mol/L, along with hexamethylenetetramine (HMTA) at a concentration of 0.0075 mol/L, both of which were dissolved in 200 mL of ultrapure water as a solvent. The substrate samples were then immersed within this mixed precursor solution and kept at a temperature of 95 °C for 5 h.

### 3.3. Fabrication of AZO/ZnO Core–Shell Nanorods by Mist CVD

After ZnO nanorods were synthesized on AZO film, a subsequent AZO coating was carried out through the mist CVD method. According to the result from our previous research [25], doping of ZnO with Al with 2% doping ratios showed a high crystalline quality and exhibited high degradation efficiency. In this research, a 2% Al doping ratio was used to form AZO/ZnO core–shell nanorods. A precursor solution was prepared by mixing ZA (0.04 mol/L) and AA (0.02 mol/L) in a solvent mixture consisting of water and methanol, with a ratio of 90 mL of water to 10 mL of methanol. Subsequently, an ultrasonic transducer at a frequency of 2.4 MHz was employed to generate mist droplets from the precursor solution within the solution chamber. These droplets, including ZA and AA, were transported to the reaction chamber by the nitrogen gas which served the dual purpose of carrier gas (flow rate: 2.5 L/min) and dilution gas (flow rate: 4.5 L/min). The substrate was set up in the reaction chamber and maintained at a deposition temperature of 400 °C. In order to investigate the effects of AZO coating on the ZnO nanorods, the coating times were set at 5, 10, 15, and 20 min. The details of the AZO coating conditions using mist CVD are summarized in Table 3.

### 3.4. Photodegradation Measurement

The photodegradation process of the obtained AZO/ZnO core–shell nanorods was performed within a glass beaker at ambient room temperature. A mixed precursor solution of MR dyes with a concentration of 1 × 10^−5^ mol/L was dissolved in the deionized water with a volume of 70 mL. Before the photocatalytic experiment, the mixed solution was wrapped with Al foil to avoid light exposure for half an hour and was adequately stirred prior to irradiation. Following this stage, the AZO/ZnO core–shell nanorods with different AZO coating times were individually immersed in the prepared MR solution and irradiated under UV light with a wavelength of 254 nm for 5 h.

### 3.5. Characterizations

The thickness of the as-deposited AZO film was measured using spectroscopic ellipsometry (WVASE32, J.A. Woollam, Co., Inc., Lincoln, CA, USA). The morphologies of the AZO film, ZnO nanorods, and AZO/ZnO core–shell nanorods were evaluated by field emission scanning electron microscopy with X-ray energy-dispersive spectroscopy (FE-SEM, SU-8020, Hitachi, Tokyo, Japan). The structural properties of obtained samples were investigated by grazing incidence X-ray diffraction (GIXRD, ATX-G, Rigaku, Tokyo, Japan). A UV-visible spectrophotometer (U-4100, Hitachi, Tokyo, Japan) was used to evaluate the optical properties of obtained samples as well as the absorption spectra of the MR solution. All the measurements were carried out at room temperature.

## 4. Conclusions

The mist CVD method was a sufficient method to coat AZO thin film on ZnO nanorods to form AZO/ZnO core–shell nanorods. The crystal growth direction of AZO layers coated on ZnO nanorods followed the same growth direction as that of ZnO nanorods which had grown along the (0001) crystal plane. The crystallinity of the AZO/ZnO core–shell nanorods was significantly enhanced as the coating time increased. All obtained AZO/ZnO core–shell nanorods showed an optical transmittance of above 55% within the visible region. Efficient photodegradation for MR dye solution using the obtained AZO/ZnO core–shell nanorods was achieved under UV light irradiation. The AZO/ZnO core–shell nanorods coated with AZO for 20 min exhibited the highest photodegradation efficiency, with a corresponding degradation reaction rate of 0.0053 min^−1^. The obtained AZO/ZnO core–shell nanorods will be significantly advancement in the degradation efficiency of MR dyes and have a high potential to be applied for industrial wastewater treatment.

## Figures and Tables

**Figure 1 molecules-29-00309-f001:**
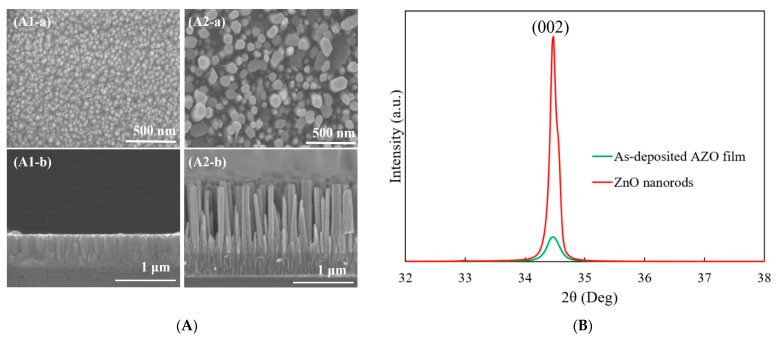
(**A**) SEM images of (**1**) as-deposited AZO film and (**2**) ZnO nanorods synthesized on AZO film: (**a**) top view and (**b**) cross-section view; (**B**) XRD patterns of as-deposited AZO film and ZnO nanorods synthesized on AZO film.

**Figure 2 molecules-29-00309-f002:**
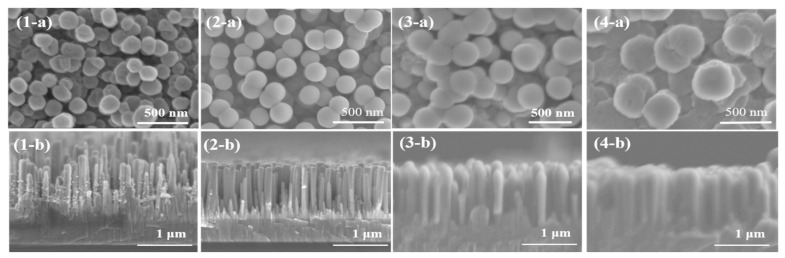
SEM images of AZO/ZnO core–shell nanorods varied with different coating times using mist CVD method ((**1**) 5 min; (**2**) 10 min; (**3**) 15 min; (**4**) 20 min; (**a**) top view; (**b**) cross-section view).

**Figure 3 molecules-29-00309-f003:**
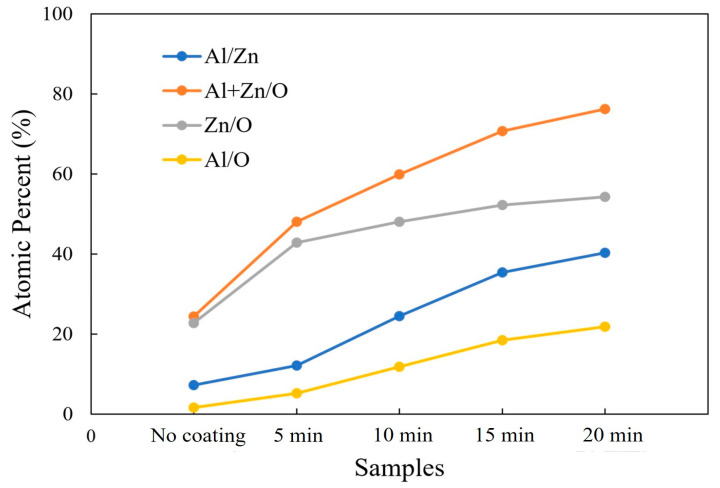
Variations in Al/Zn, (Al + Zn)/O, Zn/O, and Al/O atomic ratios calculated from EDX analysis of non-coated ZnO nanorods and AZO-coated ZnO nanorods with different coating times using the mist CVD method.

**Figure 4 molecules-29-00309-f004:**
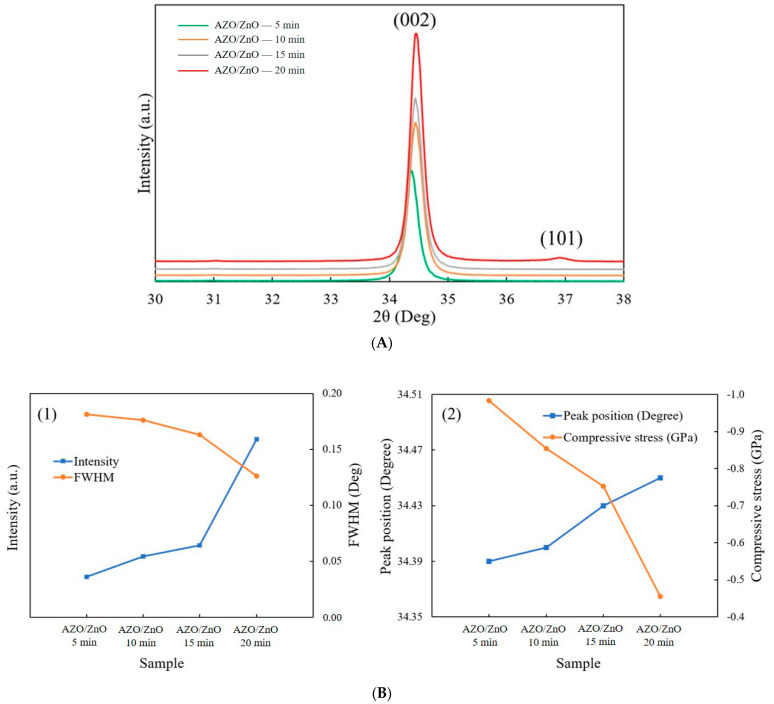
(**A**) XRD patterns of AZO/ZnO core–shell nanorods; (**B**) (**1**) the relationship of intensity and FWHM and (**2**) (002) peak position and compressive stress of AZO/ZnO core–shell nanorods varied with different coating times using the mist CVD method.

**Figure 5 molecules-29-00309-f005:**
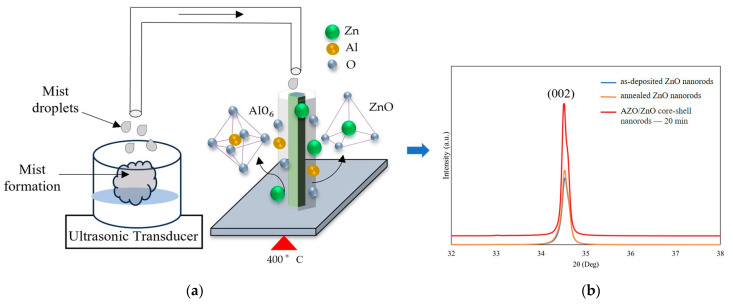
(**a**) Growth model of AZO films coated on ZnO nanorods at 400 °C during the mist CVD method; (**b**) comparison of crystallinity between as-deposited ZnO nanorods, annealed ZnO nanorods, and AZO/ZnO core–shell nanorods with 20 min-AZO coating time.

**Figure 6 molecules-29-00309-f006:**
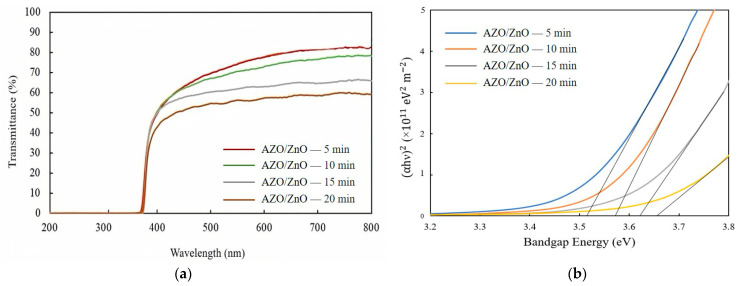
(**a**) Transmission spectra and (**b**) variation in (αhυ)^2^ of AZO/ZnO core–shell nanorods with different coating times using the mist CVD method.

**Figure 7 molecules-29-00309-f007:**
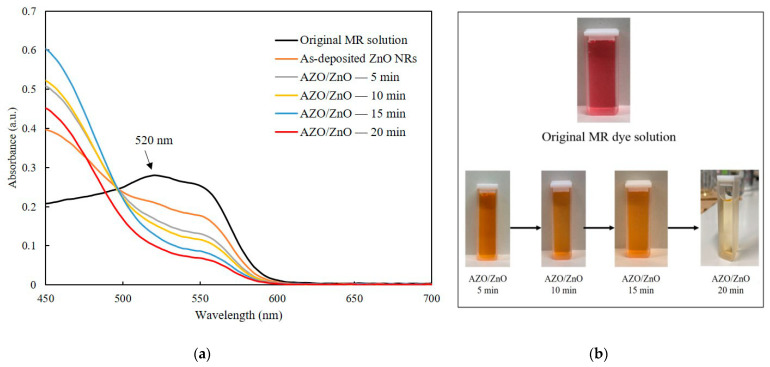
(**a**) Absorption spectra of MR solution of the as-deposited ZnO nanorods and AZO/ZnO core–shell nanorods under UV irradiation and (**b**) MR dye fading of AZO/ZnO core–shell nanorods with different coating times using the mist CVD method.

**Table 1 molecules-29-00309-t001:** Degradation rate of irradiated MR solution of AZO/ZnO core–shell nanorods under UV light irradiation.

Sample	Degradation Rate, k (min^−1^)
AZO/ZnO—5 min	0.0039
AZO/ZnO—10 min	0.0042
AZO/ZnO—15 min	0.0043
AZO/ZnO—20 min	0.0053

**Table 2 molecules-29-00309-t002:** Deposition conditions of the AZO film using RF magnetron sputtering method.

Deposition Parameter	Condition
Target	AZO (ZnO:Al_2_O_3_ = 98:2 wt%)
Substrate	Glass
Temperature (°C)	150
Power (W)	100
Pressure (Pa)	1
Working gas, Ar (sccm)	30

**Table 3 molecules-29-00309-t003:** Conditions of the AZO coating on ZnO nanorods using the mist CVD method.

Deposition Parameters	Conditions
Solute	Zinc acetate, aluminum acetylacetonate
Solvent	Methanol, water
Concentration (mol)	0.04
Doping concentration (%)	2
Substrate	ZnO nanorods/AZO thin film
Temperature (°C)	400
Carrier gas, Flow rate (L/min)	N_2_, 2.5
Dilution gas, Flow rate (L/min)	N_2_, 4.5
AZO coating time (min)	5, 10, 15, 20

## Data Availability

Data are contained within the article.

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
