# Peer review of "Synthesis of AZO-Coated ZnO Core–Shell Nanorods by Mist Chemical Vapor Deposition for Wastewater Treatment Applications"

_molecules, 2024, doi:10.3390/molecules29020309_

Round 1
Reviewer 1 Report
Comments and Suggestions for Authors
The study presents the mist chemical vapor deposition (CVD) of AZO/ZnO core-shell structures by coating AZO thin films onto ZnO nanorods. The crystal growth direction of AZO-coated nanorods was aligned with that of pure ZnO nanorods, exhibiting enhanced crystallinity as the coating time increased from 5 to 20 minutes. The crystallinity was confirmed by XRD and the transmittance of approximately 55% in the visible spectrum was achieved. Moreover, the core-shell structures demonstrated impressive efficiency in degrading MR dye solutions under UV light, with the 20-minute AZO-coated AZO/ZnO core-shell nanorods displaying the highest photodegradation efficiency and a corresponding degradation rate of 0.0053 min^-1. These findings hold promise for advancing the degradation efficiency of MR dyes. However, some arguments need to be explained and reevaluated to improve the scientific soundness of the conclusions:
1. The author explained the improved photodegradation efficiency of AZO/ZnO core-shell structures to the suppressed electron-hole recombination, which is attributed to the larger band gap increased from 3.52 eV to 3.65 eV when the coating time was increased. What is the relationship between the optical bandgap and electron-hole recombination rate? A larger band gap does not necessarily mean a lower electron-hole recombination rate. The author needs to look into other possible reasons.
2. In Figure 6a, what is the reason for the increase of absorption at around 400 nm? Although it is mentioned in the paper that the highest absorption band at 520 nm was selected as a reference, it is hard to not notice the strong peak at 400 nm and its increase with longer AZO deposition time.
3. The diameter D_AZO and D_ZNO measured from the SEM images should have some kind of distribution. Hence, the diameter ratio in Figure 3b should have an error bar related to the standard deviations of D_AZO and D_ZnO.
4. The authors demonstrate the crystallinity of the AZO/ZnO core-shells structures based on the XRD pattern. In Figure 4a, there seems to be an additional peak at 37 degrees for the 20 mins sample. Is this peak potentially assigned to other crystal planes of ZnO?
5. Some grammatical errors need to be corrected. For example, in line 66, it should be ‘resulting in enlarged bandgap energy’. On line 136, do the authors mean ‘with the volume of 70 mL’ instead of ‘with the concentration of 70 mL’?
Author Response
Dear first reviewer,
Please kindly see the attachement.
Best regards,
Htet Su Wai (First author)

Reviewer 2 Report
Comments and Suggestions for Authors
The manuscript by H.S. Wai, T. Ikuta and C. Li presents the results of depositing thin films of Al-doped zinc oxide (AZO) onto (0001)-oriented ZnO nanorods and testing them as photocatalysts for the degradation of methyl red (MR) dye in an aqueous solution.
The main scientific novelty is an application of a well-developed mist chemical vapor deposition (mist-CVD) approach to obtain AZO films onto a specific carrier, namely ZnO nanorods.
, since the used procedure is well developed for AZO. It can be assumed that in this way the efficiency of ZnO as a photocatalyst will improve due to the strong orientation of the nanorods and the presence of the AZO film. This provides a general motivation for research. However, sufficient comparisons with literature effectiveness or model objects are not provided, and some of the conclusions seem premature.
Thus, in general, the manuscript could be published in Molecules, although it appears to be more focused on Nanomaterials or Coatings. However, it needs significant improvement.
1) During describing and discussing photocatalytic studies, the authors completely ignore the literature data, although there is a known great abundance of such studies. In fact, there is not a single reference in section 3.1! This is unacceptable because it does not allow us to evaluate the effectiveness of the research at all, and thus kills all scientific significance.
2) According to the data presented, it is impossible to agree that “crystallinity was significantly enhanced” due to the formation of an AZO film. Most likely, the observed improvement is simply due to heat treatment under mist-CVD conditions. Thus, it is necessary to add data on exposure of the same samples of ZnO nanorods in a mist-CVD reactor under the same conditions as described in Table 2, but without using a solution of aluminum and zinc precursors. Such “annealed” samples should also be tested as photocatalysts. The execution and analysis of this work is of key importance and will not require significant effort from researchers.
3) There is no data on the elemental composition of the resulting material and the AZO film in particular. In fact, it is necessary to prove the presence of aluminum in the applied AZO layer. Moreover, will aluminum diffuse into the nanorod? In addition, the method and conditions used are expected to result in the presence of appreciable amounts of impurities in the films (eg, carbon impurities). The purity of the samples must be proven.
4) There is no exact data on the thicknesses of the applied AZO layers. Moreover, it is fundamentally important to strictly demonstrate the uniformity of the AZO film distribution over the height of the ZnO nanorods, and not just be limited to the model (Figure 3). Thickness uniformity is a key point when applying films to objects of complex shape, which include nanorods.
5) The nature of the diameter distribution ratios of AZO/ZnO core-shells has not been sufficiently discussed. Is the presented relationship (Figure 3b) linear? What are the error whiskers and unit error/uncertainty for this figure? So far, it seems that the presented resolution and quality of microphotographs are not enough to build such a quantitative model, because determination errors will be very significant. Thus, there is no possibility of directional control of the parameter under consideration.
Moreover, if the authors confirm the relevance of the data in Figure 3b, it is necessary to add a hypothesis about the reasons for this non-linear nature of the parameter change.
6) In fact, there is no comparison of the effect of the deposition method on crystallinity and the nature of the distribution of the AZO film over the nanorods. However, this is also a key point. It is known that atomic layer deposition has been successfully used for similar purpose.
7) The authors ignore the appearance of a new peak in the diffraction pattern after 20 min of deposition (Figure 4a).
8) The experimental part is written carelessly. There are no references to used literature methods for the preparation of nanorods and/or films. There is currently no information about the purity of the target used. The amount of aluminum acetylacetonate is missing. There is no information on the interaction of zinc and aluminum precursors in the solution used or on the absence of such interaction.
9) The choice of conditions for deposition of AZO films should be briefly justified. What determines the choice of temperature regime/precursors?
10) The authors emphasize that “The growth orientation along (0001) crystal plane of coated AZO thin film was as same as that of zinc oxide nanorods” (lines 15-16). However, it is unclear what other possibilities were expected with such an organization of the deposition process? Discuss with references.
11) Availability of data/samples is an important part of the Molecules policy. Therefore, authors should ensure that samples can be obtained rather than being limited to "not applicable" in the Data Availability Statement and Sample Availability. A possible solution would be "Samples are available upon request to the corresponding author of this work".
12) The formulation “A solution of MR dyes A solution of MR dyes with a concentration of 1×10-5 mol/L” (lines 135-136) also raises a question. What was the solvent in this solution? Or did you mean liquid pigments?
Comments on the Quality of English Language
A quality check of the English language is required starting from the introduction. In particular, there is an error in the title of the article: it is correct “core-shell structure” instead of “core-shells structure”.
Author Response
Dear second reviewer,
Please kindly see the attachment.
Best regards,
Htet Su Wai

Round 2
Reviewer 1 Report
Comments and Suggestions for Authors
The authors have addressed my comments and questions in the revision. I recommend the publication of the manuscript.